

# GC Insights: Enhancing inclusive engagement with the geosciences through art-science collaborations

Rosalie Wright[1], Kurt Jackson[2], Cécile Girardin[3], Natasha Smith[4], Lisa M. Wedding[1]

[1]The School of Geography and the Environment, University of Oxford
[2]The Jackson Foundation
[3]Nature Based Solutions initiative, University of Oxford
[4]The Oxford University Museum of Natural History

*Correspondence to*: Rosalie A. Wright (Rosalie.wright@ouce.ox.ac.uk)

**Abstract.** The environmental geosciences remain underrepresented in art-science partnerships, thereby missing valuable opportunities to enhance inclusive engagement. Here, we highlight potential pathways for scientists and artists to co-create approaches for reaching wider audiences in order to contextualise salient environmental research solutions. We synthesise lessons learned from our collective experiences as a team of scientists, artists, and exhibition officers, evaluate the potential benefits of such collaborations, and explore opportunities to enhance inclusive engagement of environmental geoscience research through art-science partnerships.

## 1 Introduction

The adverse impacts of climate change and biodiversity loss are increasingly apparent, disproportionately affecting disadvantaged and socially vulnerable populations (Arkema et al., 2013). Now is consequently a timely opportunity for engaging wider audiences in environmental geoscience as public awareness of climate and biodiversity research has also intensified (Simis et al., 2016; Lee, 2021). Effective public engagement is critical to adapting to environmental change, as this facilitates opportunities for mutual learning among the public, decision-makers, experts, and stakeholders (Rask and Worthington, 2017). Building knowledge and awareness likewise helps to empower groups most vulnerable to these risks to take action and strengthen community resilience (Khatibi et al., 2021).

Here, we focus on the opportunities created through art-science collaborations that might enable wider public engagement within the environmental geosciences. Art is broadly defined to include many forms of creative expression, including painting, photography, film, poetry, and music (Tooth et al., 2016). We scope environmental geoscience as the study of ecological and geophysical processes that influence our environment and the impacts of associated human activities. Art-science partnerships have become increasingly popular and can take many forms (Tooth et al., 2019), ranging from the more conventional 'artist as the communicator' to truly collaborative initiatives whereby projects are co-conceived, conducted and evaluated by participants across disciplines (Mould et al., 2019). The latter can serve as a platform for knowledge co-production, where both scientists and artists can meaningfully express ideas and learn from the process (Rathwell and Armitage, 2016; Macklin





and Macklin, 2019). Art-science collaborations therefore help to bridge disciplines and facilitate multi-contributor dialogue, which is necessary to address complex environmental challenges. The resulting discourse may provide an avenue to contextualise environmental science and support research that can be of real value to society. However, geoscience continues to be underrepresented in art-science collaborations in comparison with other STEM subjects (Gates, 2017; Tooth et al., 2019).

Here, we argue that art-science approaches are capable of facilitating knowledge exchange for vital biodiversity and climate change solutions. We evaluate examples of collaborative art-geoscience projects to illuminate the diversity and value of such partnerships. We conclude by synthesising experience-based recommendations to enhance inclusive engagement and wider participation in art-science partnerships.

## 2 Methods

We conducted an initial literature review of art-environmental geoscience research to guide the preliminary semi-structured interviews, which were pilot tested with researchers in the team (Kallio et al., 2016). A team of artists, scientists and exhibition officers were then identified and invited for semi-structured interviews. The direction and themes of the semi-structured interview questions were directed by the literature and preliminary interviews (Kallio et al., 2016). The objective of the semi-structured interviews was to gain a deeper understanding of art-science partnerships that have resulted in inclusive

opportunities for knowledge co-production across disciplines and audiences. Qualitative data was gathered through a set of semi-structured interview questions for representatives from each 'discipline' regarding their purpose, aims and experiences of partaking in art-science collaborations. The pre-determined questions were intended to illuminate learning outcomes of the selected projects to provide experience-based guidance for facilitating impactful art-science partnerships (Supplementary Material). Further, we collected information on art-science collaboration outcomes and evaluated the potential for impact,

specifically in relation to inclusive informal education in the environmental geosciences.

## 3 Results

### 3.1 Lessons Learned in Art-Science Partnerships

Science communication efforts have traditionally focused on the linear communication of facts (Simis et al., 2016). Otherwise known as the 'deficit model', this assumes that knowledge gaps between scientists and public understanding result from a lack

of information, resolved by one-way communication efforts. However, this approach oversimplifies relationships between knowledge, beliefs, and behaviours (Suldovsky, 2017). The deficit model may not account for audiences whose experiences and conceptions of science are often different to those of an 'expert', trained to process information objectively (Simis et al., 2016). There is a need for science communication to transition from lecturing 'matters of fact' to co-developing narratives for 'matters of concern' (Stewart and Lewis, 2017). Art can be a medium through which individuals connect with otherwise

abstract social and ecological changes, in a manner that is engaging and without trivialising the content (Locritani et al., 2020).



Art-science may therefore be a powerful tool for building trust in otherwise intangible scientific concepts and spurring discourse around socially relevant environmental science (Mach et al., 2021). Here, we highlight key reflections and lessons learned from semi-structured interviews with scientists, artists, and exhibition officers. Many artists interested in the natural world share the very same conceptual concerns as geoscientists; each would therefore greatly benefit from collaborating to
inspire interest and positive action for the natural world.

*Establishing a partnership*

- Seek insight from previous collaborations to help guide project ideation - there is no one way to participate in an art-science collaboration.

- Build relationships based on mutual understanding, trust and respect, allowing for equitable partnerships whereby
artist and scientist may learn from one another.

- Collaborate from the beginning of a project to co-develop aims and outputs, establishing shared responsibilities.

*During a partnership*

- Plan plenty of time throughout the project to critically evaluate and reflect, as a team.

- Create safe spaces for the exchange of ideas and reflective practice - be prepared to adapt and adjust throughout.

- Engage the public within the project itself (where possible) to work to address relevant needs and priorities.

*Post-partnership considerations*

- Evaluate the short- and long-term impacts of a project to reflect on the collaborative process.

- Share evaluated findings to contribute to the evidence base on best practice for art-science partnerships.

- Support the provision of training for geoscientists in positive, engaging science communication methods that are
grounded in social science research.

**3.2 Case study 1 – Connecting biodiversity and immersive art**

An art-science exhibition hosted at the Oxford University Museum of Natural History titled 'Biodiversity' featured work by contemporary artist and environmentalist, Kurt Jackson (https://www.kurtjackson.com/about/). Jackson aims to broaden his audience's experience of nature, encouraging viewers to consider how ecosystems are changing due to climate change and
human impacts through expansive mixed-media works. As he explains, "*by being aware of the life with which we share this planet we can first appreciate it, then learn to conserve it*". Though his tools and methods may differ, Jackson's intentions as an artist are greatly similar to those of a biodiversity scientist; his immersive approach to painting allows him to document and





acknowledge the complexity and fragility of the natural world. This exhibition displayed Kurt Jackson's artworks amongst the Museum's collections, showcasing the interlinkages between art, science, and natural history. Selected works were accompanied by responses from Oxford University scientists to highlight connections with research and encourage viewers to consider what biodiversity means to them. An example can be seen in Fig.1, featuring 'Taxonomy of a Cornish Foreshore' and the researcher's response as displayed. Integrating artwork with museum specimens and contemporary research created a unique environment in which visitors could both connect with the natural world in their immediate environment whilst positively engaging with research that tackles the biodiversity crisis.

## 3.3 Case study 2 – Coupling art and climate negotiations

In order to share outcomes of the recent COP26 climate negotiations (https://ukcop26.org/) in a more accessible and memorable format, artist and scientist Dr Cécile Girardin collaborated with mural painter Lisa Curtis and youth activist Arnaud Girardin-Potts to create a 4m-long mural within the COP26 negotiation zone (Fig.S2). The piece was intended to build bridges between the many activists and civil society representatives demonstrating in Glasgow and globally, and the thousands of negotiators debating within the conference centre. Such co-development of science communication is pertinent to publicly contested and politicised matters, such as biodiversity loss and climate change (Suldovsky, 2017). This mural captured the main takeaways of COP26, deploying a digestible combination of vibrant colours, shapes, and pithy statements. The dynamism of the artwork invites viewers to interpret the interconnectedness of nature, climate, and society, explore the complexities of the climate negotiations, and alludes to the key debates that shaped the COP26 talks.

## 4 Discussion and Conclusions

Here, we explored the potential benefits of art-science collaborations in the geosciences. We highlighted how sharing knowledge through the universal language of art helps to bridge backgrounds and knowledge systems. The concept of 'seeing double' (Mould et al., 2019) - through both an art and science lens - can help scientists to understand different perspectives and relations to their subject matter (Risner et al., 2019). This is critical to contextualising environmental research and addressing multifaceted social-ecological issues. Interdisciplinary approaches to sharing socially relevant geoscience will likely have positive implications for addressing science-policy implementation gaps, as fostering dialogue helps to co-develop and establish effective solutions, and further incorporate stakeholders into the scientific process (Liverman, 2008).

Stepping outside disciplines to situate data within wider contexts can therefore increase research impact. The art-science approach consequently provides a platform for reflection and knowledge exchange (Risner et al., 2019). As explored by Van Loon et al. (2020), combining artistic practice with conventional methods for building resilience to natural hazards may provide a more holistic understanding of social and ecological risks. This can lead to more comprehensive preparation and enhanced resilience to natural disasters (Van Loon et al., 2020). Further, in providing a response to Kurt Jackson's work, the researchers in Case Study 2 were encouraged to situate their science within different contexts and explain the social relevance of their



research. Such knowledge exchange is an asset in the understanding and development of effective solutions to the climate and

biodiversity crises we are facing.

Art-geoscience projects may also capacitate audiences to 'experience' landscapes and geographic concepts they have not been exposed to (Gates, 2017). This has significant implications for inclusive outreach, as place-based education facilitates relationships between scientific theories and real-world geographies. Empowering viewers to (virtually or physically) interact with subjects allows for the individual interpretation of information, instead of acting as a recipient (Mould et al., 2019). The

resulting emotional engagement with previously impalpable concepts is important in shifting public perceptions and psychological responses to environmental change (Schneider and Simon, 2014; Lee, 2021).

Enhancing inclusive engagement within the geosciences can be achieved through art-science partnerships. Our findings suggest that enabling conditions are important to create safe spaces for the knowledge exchange and reflective practice. Starting with relationship building that is based on mutual respect was found to support the successful development of equitable

partnerships and co-production of ideas. Further, our lessons learned underscored that impact should be monitored with specific metrics that evaluate both the process of interdisciplinary collaboration and the short- and long-term impacts of an inclusive engagement project.



**Author contributions**

RW conceived the article and led the writing process, supported by LW. KJ and NS designed and ran the exhibition. CG designed and co-created the COP26 piece. All authors contributed to the development of the understanding and ideas presented.

**Competing interests**

The authors declare that they have no conflict of interest.

**Financial support**

The work was supported by the University of Oxford, ESRC IAA Knowledge Exchange Dialogues Award.

**Acknowledgements**

We thank Poppy Menzies Walker for her encouragement and sage advice on this paper.

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




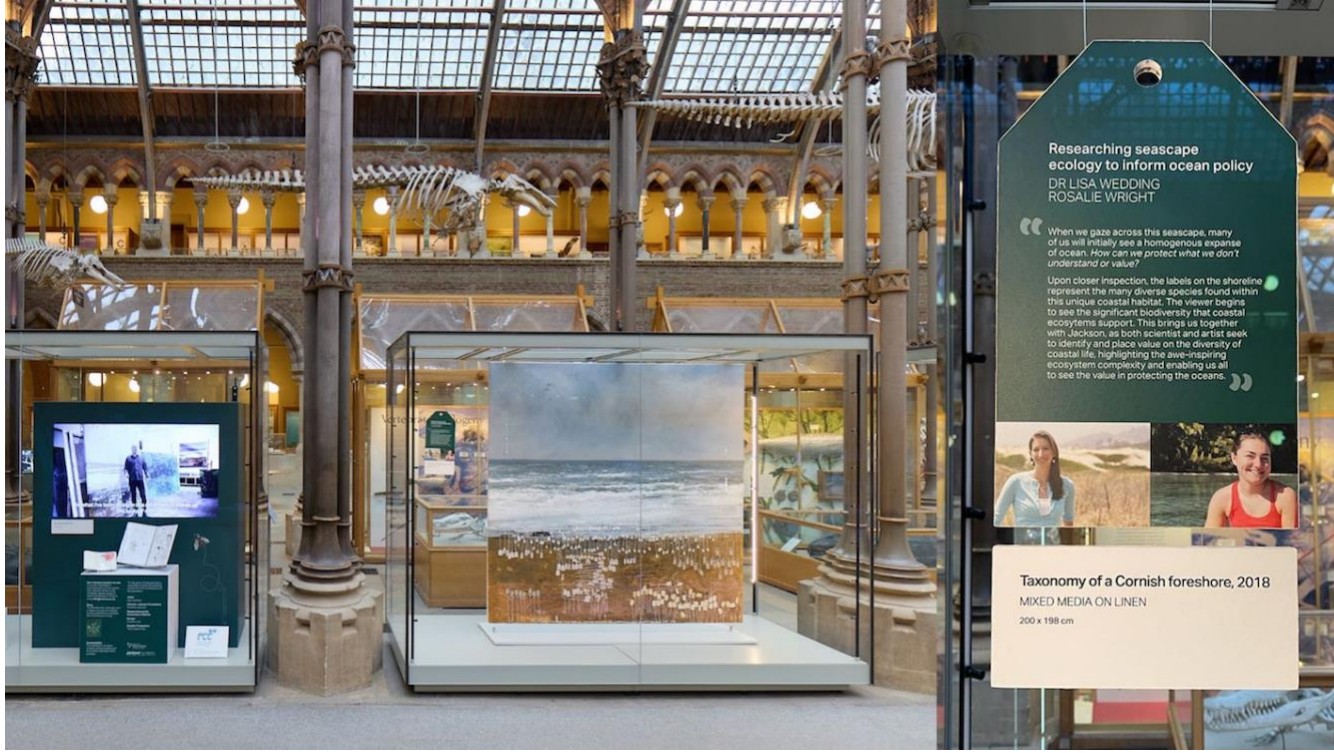

**Figure 1:** 'Taxonomy of a Cornish Foreshore' by Kurt Jackson, on display as part of the Biodiversity exhibition at the Oxford University Museum of Natural History. The piece shows how the beach, the foreshore, has a particular resonance to many whilst also being a biodiversity hotspot, a liminal zone and the meeting point for ecosystems. As Jackson explains, "*a coastline is the front line where our impact is tangible and alarmingly visible - if we allow ourselves a moment we can see and be aware of the fragility, diversity and complexity of this world, but crucially also the beauty*". This work was on display featuring the following response from authors Rosalie Wright and Dr Lisa Wedding: "*When we gaze across this seascape, many of us will initially see a homogenous expanse of ocean. How can we protect what we don't understand or value? Upon closer inspection, the labels on the shoreline represent the many diverse species found within this unique coastal habitat. The viewer begins to see the significant biodiversity that coastal habitats support. This brings us together, as both scientist and artist seeking to identify and place value on the diversity of coastal life, highlighting the awe-inspiring ecosystem complexity and enabling us all to see the value in protecting the oceans.*" (Image credit: Museum of Natural History by Ian Wallman, pixieset.com)