# Peer review of "GC Insights: Enhancing inclusive engagement with the geosciences through art-science collaborations"

_EGUsphere, 2022_

## Referee Comment (RC1)

Gc. 2022-357

*GC Insights: Enhancing inclusive engagement with the geosciences through art-science collaborations*

Wright et al.

The research presented in this GC Insight is conducted with a limited sample: a small group of participants. This is not immediately clear from the article, and become evident only in the supplement. In the end, the semi-structured interviews are conducted with three participants (an artist, an exhibition officer and a scientist/artist). Nevertheless, the lesson learned, obtained also with the help of the other few participants, are interesting and can provide a useful guideline for art-science collaborations in the environmental geosciences.

The other important aspect comes out from the case studies, and I believe it is not enough emphasized in the article: integrating Art in "unusual" context as can be a Museum of Natural History (case #1) or in a political context (case #2) can contribute in engaging with geosciences in a powerful way? (It is inspiring, for instance, what Kurt Jackson answer to *how do people respond to the works*: "And for 'Biodiversity', we get some lovely responses from the audiences. They've been sending things that they've made themselves; it never occurred to me that audiences might respond in this way.")

I believe this aspect is worth to be explored more in depth. So if the authors have elements to develop more this aspect I would suggest them to organize the article around it.  (see for instance Natasha Smith answer to *What are the key reflections and takeaways from the 'Biodiversity' exhibition?)*

**Specific comments:**

10-11 I suggest: "Here we present two cases studies as examples of how co-creating approaches for reaching wider audiences…"

41-42 Please make immediately clear that people interviewed are three.

66-80 This is a repetition of what already summarized in the supplement

98 (Fig. S2 in the supplement)

Par 4 should be reorganized. Rather than being a collection of references to other works, it should summarize and discuss what your team has achieved in collaborating and co-creating, what can be further explored, and the limits of your work (if there are).

---

## Author Response (AR1)

Rosalie A. Wright
Researcher
School of Geography and the Environment (SoGE)
University of Oxford, SoGE, South Parks Road,
Oxford, UK, OX1 3QY
Email: rosalie.wright@ouce.ox.ac.uk

January 18, 2023

Re: Revised manuscript submission to *Geoscience Communication*

Dear John,

Please find attached the updated manuscript entitled *"GC Insights: Enhancing inclusive engagement with the geosciences through art-science collaborations"*. The feedback provided by each of the two external reviewers has been particularly helpful to receive. I have revised the manuscript accordingly and made the following major revisions noted in a point-by-point response below, in relation to each comment.

**Reviewer 1 Comments:**

***The research presented in this GC Insight is conducted with a limited sample: a small group of participants. This is not immediately clear from the article, and become evident only in the supplement.***

- We have added text to make the sample size of interviews clear throughout the article. The case studies also have been moved to the Methods section to provide further context to the interviews.

***The other important aspect comes out from the case studies, and I believe it is not enough emphasized in the article: integrating Art in "unusual" context as can be a Museum of Natural History (case #1) or in a political context (case #2) can contribute in engaging with geosciences in a powerful way? I believe this aspect is worth to be explored more in depth.***

- We also believe that these more "unusual" contexts are important to highlight and explore. We have included new text to further highlight the specific values of the Museum context in our results and discussion.

***10-11 I suggest: "Here we present two cases studies as examples of how co-creating approaches for reaching wider audiences..."***

- Thank you for the suggestion, we have revised this line accordingly.

***41-42 Please make immediately clear that people interviewed are three.***

- We have clarified the interview sample size earlier and throughout the article.

***66-80 This is a repetition of what already summarized in the supplement 98 (Fig. S2 in the supplement)***

- We have clarified the interview sample size earlier and throughout the article. Lines 66 to 80 were replaced with more detailed text to remove repetitions of what is included in the supplement, and further explicate the results of each interview.

***Part 4 should be reorganized. Rather than being a collection of references to other works, it should summarize and discuss what your team has achieved in collaborating and co-creating, what can be further explored, and the limits of your work (if there are).***

- To address this comment, Part 4 has been re-written to specifically summarise and reflect upon our work, in addition to the inclusion of text addressing the limitations of this work and stating possible next steps.

**Reviewer 2 Comments:**

***The introduction draws from only a small number of examples of art-science collaboration, hence does not seem to recognise the wide breadth of activity that has been occurring. Indeed, this journal has an entire Special Issue of examples (https://gc.copernicus.org/articles/special_issue1046.html), which are largely uncited in the manuscript. This seems like a massive oversight to this reviewer and***

- We have cited several articles from this Special Issue and agree that there is a wide variety of art-science collaborations occurring in this space. We have added further comment and references (in both the introduction and discussion sections) to the Geoscience Communications Special Issue, "*Five years of Earth sciences and art at the EGU (2015–2019)*" to better reflect the breadth of art-geoscience activity and publications on this topic.

***The research aim of the article is not sufficiently well-stated to prepare readers to comprehend the manuscript. At times it appears that the authors wish to demonstrate the efficacy of art-science collaborations in co-creating knowledge and/or communicating important geoscience concepts to nontradiational audiences. At others, it seems they want to identify recommendations on how to enable successful collaborations between stakeholders in art-science collaborations (artists, scientists, exhibition staff). It is my opinion that the work cannot achieve the former, but may be able to demonstrate examples of the latter, though they will not be exhaustive due to the limited scope of the study. This should be much clearer in the abstract, introduction, and conclusions.***

- We have adjusted the abstract, introduction and conclusions to specify that our research aim was to provide reflections and recommendations on successful partnerships. We have revised text accordingly throughout the article to clarify this.

***The methods employed, as written, are rather unclear in the article. The fact that all results are drawn from just two case study projects only becomes apparent near the end of the manuscript. Exactly what the authors were looking to extract from their literature review and interview question development isn't stated - there are many vagueties such as simply stating "learning outcomes" and "information". How the interview data was processed to arrive at themes and conclusions is stated nowhere in the manuscript.***

- We have made clear throughout the article the sample size for interviews and the number of case studies. The case studies were moved to follow the Methods section to clarify this link earlier. The research aim for the literature review and interviews has been more clearly stated in the introduction, as discussed above. Due to the short nature of a GC Insights piece, we had not added further detail regarding the interviews. However we have since revised the Methods section to include further information on our scope and intentions for both the literature review and interviews.

*The results, which are presented only as bullet point recommendations, are not adequately discussed. Their relation to the underlying data is not explained and no trends, even in which groups of stakeholders they originate, are stated. The discussions and conclusions are finally largely disconnected from the research activity, the interviews.*

- We reconfigured the manuscript results and discussion sections to better reflect upon the results of the interviews, relating findings directly to the stakeholder group they originated from. The bullet point recommendations have been removed and will be provided in the Supplementary Material for further information. Further, we re-wrote the discussion to provide a more specific reflection on how the interview findings and case studies fed into the key recommendations, also acknowledging limitations to the work and possible next steps.

**Specific comments**

*L9 & 34: The claim that "environmental geosciences remain underrepresented in art-science partnerships" is not substantiated in the references provided, which correspond to a case study project and an editorial perspective. Only with a full survey or meta-analysis could this claim be substantiated, which does not appear to have occurred. Therefore, the authors should remove this claim as it misrepresents the work currently being done.*

- The claim of underrepresentation was reflective of the Tooth et al. (2019) piece for a contemporary Art-Geoscience Special Issue that drew from the Tooth et al. (2016) article in Earth Surface Processes and Landforms. However, we acknowledge that this article for Geoscience Communication was not a full survey or meta-analysis and cannot fully substantiate this, so this has been rephrased and the claim removed.

*Section 2: Overall this section is unclear, but also repetitive in many aspects. It could do with rewriting with a clear focus. Examples of unclear aspects are: What you were looking to find in literature review? How many people were identified to interview? What was the makeup? What was the criteria in identifying?; Who the "outcomes and impact" are meant to be on? If the stakeholders interviewed and their partners then that would be appropriate, however, if this is meant to concern the attendees then I do not agree that the study can adequately assess these.*

- Based on this feedback, we have rewritten the Methods section to be more concise and added detail regarding specific aims and criteria for both the literature review and interviews.

*Case studies: These should be stated outright before any results are presented, so it provides much needed context to what the interview data is in relation to - their experiences of undertaking these two projects.*

- To address this comment, the case studies have been moved to precede the results section and provide further context regarding the interview data. We have included additional text to clarify this link in the introduction, methods and results sections.

*Lines 53-65: These read like more introductory material unrelated to any results of the study, hence should be moved accordingly.*

- These lines summarise findings of our literature review, though we acknowledge they are better suited to the introduction. We have moved the most relevant content to the introduction.

Thank you again to the reviewers and for your guidance on this manuscript. It is much appreciated, and we look forward to hearing from you. Please let us know if you have any additional questions and thank you for the opportunity to submit our work for your consideration.

Sincerely,

Rosalie Wright

Rosalie A. Wright,
Seascape Ecology Lab
School of Geography and the Environment
University of Oxford

---

## Author Response (AR2)

Rosalie A. Wright
Researcher
School of Geography and the Environment (SoGE)
University of Oxford, SoGE, South Parks Road,
Oxford, UK, OX1 3QY
Email: rosalie.wright@ouce.ox.ac.uk

[Figure]

[Figure]

February 28, 2023

Re: Revised manuscript submission to *Geoscience Communication*

Dear John,

Please find attached the updated manuscript entitled *"GC Insights: Enhancing inclusive engagement with the geosciences through art-science collaborations"*. I have revised the manuscript and made the following minor revisions noted in a point-by-point response below, in relation to each comment.

**Reviewer 1 Comments:**

*Just a minor suggestion: since the aim of your study is to evaluate two retrospective case-studies, it would be appropriate indicating the years in which the two art-science collaborations took place.*

- We have added dates to each of the case study summaries.

**Reviewer 2 Comments:**

*My suggestion would be to use a two panel Figure with sublabels to include both case studies in the manuscript, i.e. integrating both figures in the Supplement. I would also argue that the current Figure caption is beyond a simple description of what is shown in the Figure, hence if the additional text (e.g. quotes) are required these may perhaps be better integrated into the main text or supplement.*

- We have updated the figure to include labelled panels of each case study image. We reduced the caption text to provide a simple description of each image, as requested.

Thank you again to the reviewers and for your guidance on this manuscript. It is much appreciated and thank you for the opportunity to submit our work for your consideration.

Sincerely,

Rosalie Wright

Rosalie A. Wright,
Seascape Ecology Lab
School of Geography and the Environment
University of Oxford